# A study on the association between eye movements and regular mouthing movements (RMMs) in normal fetuses between 24 to 39 weeks of gestation

**Kana Maehara**[1], **Seiichi Morokuma**[2]*, **Kazushige Nakahara**[1], **Hikohiro Okawa**[1], **Kiyoko Kato**[1]

1 Department of Obstetrics and Gynecology, Graduate School of Medical Sciences, Kyushu University, Fukuoka, Japan, 2 Department of Health Sciences, Graduate School of Medical Sciences, Kyushu University, Fukuoka, Japan

* morokuma@med.kyushu-u.ac.jp

**Data Availability Statement:** All relevant data are within the manuscript.

## Abstract

Regular Mouthing Movements (RMMs) are movements in which lips and lower jaw movements occur regularly and can be observed in the fetus using transabdominal ultrasonic tomography. In near term infants, it is known that RMMs form clusters during the quiet sleep period. The notation of RMMs is not uniform, and is described as spontaneous sucking movement or non-nutritive sucking in newborns. Non-nutritive sucking is used to evaluate neurological function after birth, but there are no fetal indicators. The purpose of this study was to clarify the changes in the RMM clusters in fetuses at 24–39 weeks of gestation, and to investigate the relationship with the non-eye movement (NEM) period, which corresponds to the quiet sleep period after birth. Subjects included 83 normal single pregnancy cases. Fetal RMMs and eye movement (EM) were observed for 60 minutes using ultrasonic tomography and recorded as moving image files. We created time series data of eye movements and mouth movements from video recordings, and calculated RMM clusters per minute within effective observation time, RMM clusters per minute in EM period, RMM clusters per minute in NEM period, mouthing movements per cluster and ratio of number of RMM clusters per minute between NEM and EM periods and analyzed using linear regression analysis. As a result, critical points were detected in at two time points, at 32–33 weeks and 36–37 weeks of gestation, in RMM clusters per minute within the effective observation time and RMM clusters per minute in NEM period, respectively. RMM clusters in human fetuses increased from 32–33 to 36–37 weeks. This change is thought to represent fetal sleep development and central nervous system development.

## Introduction

Regular Mouthing Movements (RMMs) are frequent movements of the lips and lower jaw and can be observed by imaging the fetal face using ultrasonic tomography [1]. In near-term infants, it is known that RMMs form clusters during quiet sleep [2,3,4].

**Funding:** SM received each award. This study was supported by a research grant from the Japan Society for the Promotion of Science KAKENHI (grant no.: 16H01880, 16K13072, 18H00994, 18H03388), AMED under Grant Number 19gk0110043h0001 and RIKEN Healthcare and Medical Data Platform Project. The funders had no role in study design, data collection and analysis, decision to publish, or preparation of the manuscript.

**Competing interests:** The authors have declared that no competing interests exist.

The nomenclature for RMMs is non-uniform and may differ between fetuses and newborns. In neonates, the terms "Spontaneous sucking movements" and "Non-Nutritive Sucking (NNS)" are commonly used. Both of these represent movements that appear as a cluster of 2 to 4 seconds of spontaneous regular lip movements, similar to RMMs [2,4,5]. NNS and RMMs are considered to be the same in fetuses and newborns due to their periodicity and appearance pattern [2,4,5].

A study of full-term human infants has reported that RMM clusters appear in association with high-amplitude slow waves that are characteristic of non-REM sleep [6]. In animal experiments using rats, it has been reported that the development of slow waves is associated with synaptic plasticity and cerebral cortex maturation [7,8]. Based on the above, it is considered that RMM clusters are related to the development of brain functions related to sleep.

In previous studies on fetal mouth movements, it has been reported that short-interval mouth movements increase at 32–34 gestational weeks [9,10] while short-interval mouth movements during non-eye movement (NEM) periods increase at 35 weeks of gestation [9], exploring interval times of all mouth movements within the observation time. Furthermore, E. E.van Woerden reported that RMMs were observed in the NEM period in 74% cases at 38–40 weeks of gestation [5]. In addition, recognizing RMMs during the NEM period has been used as an index for evaluating fetal central nervous system (CNS) function [11,12]. The interval time of mouth movements and the presence or absence of RMMs have been studied only for the last trimester have been studied, but there are no reports on the mode of occurrence of RMM clusters over longer time points. Furthermore, due to the relationship with slow waves, RMMs need to be studied in relation to cluster formation. By investigating fetal RMM clusters and the relationship between RMM clusters and the NEM period, it is possible to identify indicators of fetal CNS function development and neurological prognosis. By observing eye movements of the fetus using ultrasonic tomography, the eye movement (EM) and NEM periods are recognized and they are considered to correspond to REM and non-REM sleep after birth. The NEM period continues from around 24 gestational weeks [13,14,15].

Based on the above, the purpose of this study was to clarify changes in RMM clusters in fetuses between 24–39 gestational weeks and to investigate the relevance to the NEM period.

## Materials and methods

### Fetal population

The name of ethics committee is ethics committee at Kyushu University Hospital. The study was approved by our ethics committee (No. 27–51) and informed consent was written from all mothers prior to the start of the study. We performed a cross-sectional study of 101 normal singleton pregnancies between 24 and 39 weeks' gestation that underwent perinatal management at the Maternity and Prenatal Care Unit of Kyushu University Hospital from 2013 to 2019. Cases with apparent fetal morphological abnormalities and maternal complications at the time of recruitment were excluded. However, after data collection, as the pregnancy progressed, there were 5 cases diagnosed with gestational diabetes mellitus and 5 cases diagnosed with hypertensive disorders of pregnancy. We calculated the time since the last menstrual period and determined the number of gestational weeks in the first trimester by measuring crown-rump length using ultrasonic tomography. In all cases, the mothers had no history of smoking or alcohol intake. They also did not take any medications other than iron or vitamins during pregnancy. There were no particular complications during labor and no developmental abnormalities at the one-month infant checkup.

## Data acquisition

Patients were placed in a supine position in a quiet room, allowing them to change positions freely. The procedure was performed between 13:00 and 16:00 at least 2 hours after meal intake. Fetal eye movements and mouth movements were observed for 60 minutes at a frame rate of 30 frames/s or higher using transabdominal two-dimensional sonography (APLIO 500 TUS-A500; TOSHIBA, Japan) with a 3.5MHz convex transducer (PVT375BT Probe). The video data was saved on an SD card as a digital video file in MP4 format. A cross section of fetal eye movements and mouthing movements were seen on coronal imaging in which the edge of the fetal lens was depicted as a ring-shaped circular echoic image at the same time that the mouth was observed (S1 Fig). When we were unable to render the appropriate cross-section, we adjusted the position of the probe and asked the mother to change her position so that the proper cross-section could be rendered.

## Analytical methods

**Data processing.**   For each eye movement and mouth movement, we created time-series data using saved videos [16]. Next, we divided every minute, and the periods during which eye movements occurred were defined as EM periods, while those during which eye movements did not occur were defined as NEM period. The time series data of mouth movements were taped at the beginning of the mouth movement and considered as one mouth movement. The time between one mouth movement and the next mouth movement was defined as the interval time between mouth movements. Moreover, mouth movements that occurred two or more times with less than one second interval in between were defined as RMM clusters [1]. We defined the period of observation as the effective observation time, and examined the data within the effective observation time. There are times when the eye and mouth movement of the fetus cannot be observed due to the movement of the fetus. If no facial movements can be observed, the time during which no observations were made was excluded and the analysis was performed on the effective observation time. In addition, we analyzed cases where the effective observation time was 80% or more.

The items used as indicators in the analysis are as follows.

1. Effective observation time (min): Time during which mouth movements were identified during the observation period. Cases in which effective observation time was 80% or more (48 minutes or more) of observation time were analyzed.

2. Total number of RMM clusters: Total number of RMM clusters observed within the effective observation time.

3. Total number of RMM clusters in EM: Total number of RMM clusters observed during the EM period.

4. Total number of RMM clusters in NEM: Total number of RMM clusters observed during NEM period.

5. RMM clusters per minute = Total number of RMM clusters/effective observation time

6. RMM clusters per minute in EM = Total number of RMM clusters/EM period (min)

7. RMM clusters per minute in NEM = Total number of RMM clusters/NEM period (min)

8. MMs per cluster = Total number of mouth movements in RMM clusters/total number of RMM clusters (min)

9. Ratio of number of RMM clusters per minute between NEM and EM periods = RMM clusters per minute in NEM/ RMM clusters per minute in EM

**Fetal developmental groups.** Cases between 24–39 weeks' gestation were classified into eight groups of 2-week intervals. (24–25 weeks, 26–27 weeks, 28–29 weeks, 30–31 weeks, 32–33 weeks, 34–35 weeks, 36–37 weeks, and 38–39 weeks).

**Piecewise linear regression analysis.** In the scatter plot of each variable, analysis was performed using a piecewise linear regression model to identify critical points between 24–25 and 38–39 gestational weeks [17, 18]. To select the best regression equation, Mallows' $Cp$ value was defined with the equation $Cp = RSS / s^2 - (n-2p)$. In this equation, $n$ was the number of fetuses, $p$, the number of critical points, $RSS$, the residual sum of the square from a given combination of $p$ points, while $s^2$ was the residual mean square based on regression using all points [19, 20]. The optimal piecewise linear regression was selected by two steps. At first, $p$ was determined as the smallest $p$ for which the $Cp$ value was the minimum value less than or equal to $p$. Second, among the found combinations of $Cp$ values, the minimum combination of $Cp$ values was selected [19,20]. In this analysis method, both end groups (24–25 and 38–39 weeks) were excluded as they might be detected as "critical points". For each index, statistical analysis was performed using Student's t-test in the first group, the last group, and the "critical points" group. All analyses were performed using R 3.2.5 statistical software (https://www.r-project.org/).

**Verification of the reliability of the taping process.** The reliability of the taping method has been verified in a paper by Okawa et al. The intraclass correlation coefficient (ICC) in the taping method was ICC> 0.8, which is considered to reflect good reliability [16].

## Results

## Experimental outcomes and analyses

In two cases, the effective observation time was insufficient due to fetal posture and movement. These two cases were observed at gestational weeks 36 and 38 weeks, respectively. Eight cases of small-for-gestational age (SGA) infants (infants with birth weight less than the tenth percentile by birth standard value by gestation period), one case which had no NEM period during the effective observation time, and seven cases with no RMM clusters within the effective observation time were excluded. Analysis was performed in 83 out of 101 cases. The clinical characteristics of analyzed cases are shown in Table 1.

**Table 1. Characteristics of the 83 fetuses.**

| Age group (weeks) | n | Gestational age at delivery (weeks.days) | Birth weight (g) | Sex (male/female; n) | Apgar score | | pH of the umbilical artery |
|---|---|---|---|---|---|---|---|
| | | | | | 1 min | 5 min | |
| 24–25 | 10 | 39.3 (37.5–40.6) | 3011 (2745–3670) | 5/5 | 8 (7–9) | 9 (8–10) | 7.27 (7.19–7.36) |
| 26–27 | 9 | 39.1 (36.3–40.6) | 3050 (2565–3330) | 4/5 | 8 (7–9) | 9 (8–10) | 7.30 (7.27–7.35) |
| 28–29 | 10 | 38.3 (36.0–40.1) | 3180 (2340–3175) | 5/5 | 8 (8–9) | 9 (9–10) | 7.28 (7.25–7.35) |
| 30–31 | 12 | 39.6 (37.4–42.1) | 3240 (2340–3650) | 6/6 | 8 (8–9) | 9 (9–10) | 7.33 (7.24–7.44) |
| 32–33 | 10 | 38.5 (37.3–41.1) | 2930 (2545–3135) | 4/6 | 8 (4–9) | 9 (7–10) | 7.30 (7.22–7.38) |
| 34–35 | 11 | 38.6 (38.1–40.5) | 3020 (2600–3400) | 6/5 | 8 (7–9) | 9 (9–10) | 7.26 (7.22–7.39) |
| 36–37 | 10 | 39.5 (38.3–41.0) | 3228 (2770–3885) | 7/3 | 8 (7–9) | 9 (9–10) | 7.23 (7.12–7.39) |
| 38–39 | 11 | 40.1 (38.6–41.3) | 3095 (2765–3828) | 7/4 | 9 (8–9) | 9 (8–10) | 7.32 (7.28–7.35) |
| Total | 83 | 39.2 (36.0–42.1) | 3075 (2340–3885) | 44/39 | 8 (4–9) | 9 (7–10) | 7.30 (7.12–7.44) |

Data is shown as medians with ranges.

**Table 2. Effective observation time and rate of eye movement (EM) and non-eye movement (NEM).**

| Age group (weeks) | Effective observation time (min) | EM period rate (%) | NEM period rate (%) |
|---|---|---|---|
| 24–25 | 59 ± 2 | 56 ± 13 | 47 ± 14 |
| 26–27 | 59 ± 3 | 60 ± 16 | 41 ± 15 |
| 28–29 | 60 ± 2 | 65 ± 12 | 37 ± 12 |
| 30–31 | 59 ± 3 | 68 ± 11 | 32 ± 12 |
| 32–33 | 59 ± 2 | 68 ± 7 | 34 ± 6 |
| 34–35 | 58 ± 2 | 63 ± 8 | 35 ± 8 |
| 36–37 | 59 ± 2 | 60 ± 13 | 44 ± 16 |
| 38–39 | 59 ± 1 | 65 ± 12 | 35 ± 13 |

Data is represented as means ± standard deviation.

There were no significant differences between groups.

Table 2 shows the effective observation time, EM time (%), and NEM time (%) of each 2-week group. Table 3 shows the number of cases in which RMM clusters were observed within the effective observation time, and the number of cases in which RMM clusters were observed in the NEM period.

The results of Piecewise linear regression analysis are shown below. RMM clusters per minute had two statistically significant critical points at 32–33 weeks and 36–37 weeks' gestation ($Cp$ = -0.13). RMM clusters per minute did not increase from 24–25 gestational weeks to the critical point at 32–33 gestational weeks ($p$ = 0.48), but increased significantly after 32–33 gestational weeks. ($p$ = 0.01). There was no significant difference in RMM clusters per minute between 36–37 weeks and 38–39 weeks gestation ($p$ = 0.07) (S2 Fig).

RMM clusters per minute in EM had statistically critical points at 26–27 weeks of gestation ($Cp$ = -0.05). However, there was no significant difference in either 24–25 weeks and 26–27 weeks of gestation or 26–27 weeks and 38–39 weeks of gestation ($p$ > 0.05) (S3 Fig).

RMM clusters per minute in NEM had two statistically significant critical points at 32–33 weeks and 36–37 weeks of gestation ($Cp$ = 0.32). RMM clusters per minute in NEM did not increase from 24–25 gestational weeks to the critical point at 32–33 gestational weeks ($p$ = 0.65), but increased significantly after 32–33 gestational weeks. ($p$ = 0.03). There was no significant difference in RMM clusters per minute between 36–37 weeks and 38–39 weeks of gestation ($p$ = 0.22) (S4 Fig).

**Table 3. Presence or absence of RMMs in each 2-week group in the effective observation time and NEM period respectively.**

| Age group (weeks) | n | Presence of RMMs in effective observational time | Presence of RMMs in NEM period |
|---|---|---|---|
| 24–25 | 12 | 10 (83) | 9 (75) |
| 26–27 | 11 | 9 (82) | 6 (55) |
| 28–29 | 11 | 10 (91) | 6 (55) |
| 30–31 | 12 | 12 (100) | 8 (67) |
| 32–33 | 11 | 10 (91) | 8 (73) |
| 34–35 | 11 | 11 (100) | 8 (73) |
| 36–37 | 11 | 10 (91) | 9 (82) |
| 38–39 | 11 | 11 (100) | 9 (82) |

Values in parentheses represent percentages.

MMs per cluster had statistically critical points at 26–27 weeks of gestation ($Cp$ = 0.81). However, there was no significant difference in either 24–25 weeks and 26–27 weeks of gestation or 26–27 weeks and 38–39 weeks of gestation ($p > 0.05$) (S5 Fig).

The ratio of the number of RMM clusters per minute between NEM and EM periods had statistically critical points at 36–37 weeks of gestation ($Cp$ = -2.85). However, there was no significant difference at either 24–25 weeks and 36–37 weeks of gestation or 36–37 weeks and 38–39 weeks of gestation ($p > 0.05$) (S6 Fig).

## Discussion

### Main findings

Both RMM clusters per minute in EM and RMM clusters per minute in NEM had critical points at 32 to 33 weeks and at 36 to 37 weeks of gestation, and a significant increase was observed between the critical points. This suggests that the changes in the occurrence of RMM clusters were related to the occurrence of RMM clusters synchronized to the NEM period.

### Comparison with existing literature

According to van Woerden et al., RMM clusters were observed in 74% of fetuses between 38 to 40 gestational weeks at 1F corresponding to quiet sleep in the neonatal period [5,21,22]. In addition, among targeted low-risk fetuses after 36 gestational weeks, Pilai et al. reported that 81.8% of cases had RMM clusters at 1F, 6.8% had no RMMs at 1F, and in 11.4% RMM clusters were not ascertained during the effective observation time [23]. In this study, among fetuses greater than 36 gestational weeks, 18 cases (81.8%) had RMM clusters in the NEM period, and 3 cases (13.6%) had no RMM clusters observed during the NEM period. In one case (4.5%) no RMM clusters were observed within the effective observation time. This is in keeping with previous reports, where it was found that there were cases that with no RMM clusters observed at a certain rate.

Horimoto et al. reported the cumulative incidence of mouth movements at every time interval [9]. The cumulative incidence of mouth movements at every time interval between 28–31 weeks of gestation showed no bias. However, there was a bias concerning mouth movements with interval times less than 1 second at 35–41 weeks of gestation. Cases between 32–34 gestational weeks had a transition period between the two weeks. In this study, RMM clusters did not increase or decrease from 32 to 33 weeks' gestation but increased significantly from 32 to 33 weeks to 36 to 37 weeks of gestation. The transition period shown in the previous study may have been due to an increase in RMM clusters.

Pineda et al. measured NNS per burst and NNS bursts per minute between 32 and 43 weeks of PMA (post-menstrual age) in preterm infants born before 32 weeks' gestation. It had been reported that both indicators increase with advancement in PMA [24]. In this study, RMM clusters per minute equivalent to NNS per burst per minute increased from 32 to 37 weeks of gestation, but MMs per cluster equivalent to NNS per burst did not change with advancement of gestational age. In the study by Pineda et al., the median NNS per burst at PMA of 32 to 39 weeks was not significantly increased or decreased, and the results were similar in this study. Moreover, other studies of preterm infants have reported that poor NNS is an indicator of CNS dysfunction [25,26]. These studies show that understanding changes in NNS is useful for assessing normal development of the CNS.

In a fetal study, it was reported that the secondary sulcus develops rapidly and the cerebral wall increases remarkably from 29 weeks to 34 weeks' gestation [27]. NREM sleep also has been reported to begin to appear between 33–35 weeks of gestation when neuronal

connections in the thalamo-cortical region and brainstem begin to function [28]. This coincides with the time when the number of RMM clusters in the NEM period increased in this study.

In a study of full-term neonates, there was an association between high-amplitude electroencephalograph waves during non-REM sleep and RMM clusters. The slow rhythm waves of the electroencephalograph during non-REM sleep were recorded when the majority of cortical neurons in a specific brain region were involved [8]. It is generally believed that slow waves originate in the neocortex [29,30,31]. Based on the above findings, it might be considered that the changes in RMM clusters observed in this study represent fetal sleep development and CNS development.

By continuously observing the infant after birth, it may be possible to confirm the developmental prognosis of the baby in relation to RMM. Therefore, there is a possibility that these data can be clinically used as one of the indicators of neurodevelopment in the future.

## Strengths and limitations

This is the most detailed study of RMMs in human fetuses. This study has two limitations. The first limitation is due to the method of observing the fetus with ultrasound. Cases where such observation was not possible, or in which the effective observation time was insufficient were not included, which may have affected the results of this study. The second limitation is that the duration of the study was limited to only the gestational period and no comparison was made to postnatal neurological prognosis. Furthermore, as the number of cases included in the analysis was limited, future studies should use larger sample sizes. Despite the above limitations, a certain tendency was observed in the development process of RMMs.

## Conclusion

RMM clusters per minute in NEM increased from 32–33 weeks to 36–37 weeks of gestation. These results may be indicative of developments in the CNS and fetal sleep mechanism.

## Supporting information

**S1 Fig. Cross-sectional observation by sonography.** A cross section of the fetal eye movements and mouthing movements observable on coronal imaging in which the edge of the fetal lens was depicted as a ring-shaped circular echoic image at the same time that the mouth was observed.
(TIF)

**S2 Fig. Bar graph of RMM clusters per minute.** The horizontal axis indicates the RMM clusters per minute and the vertical axis indicates the weeks of gestation. The bars represent means and standard deviations. $^*p < 0.05$, n.s.; not significant.
(TIF)

**S3 Fig. Bar graph of RMM clusters per minute in EM.** The horizontal axis indicates RMM clusters per minute in EM and the vertical axis indicates gestational weeks. The bars represent means and standard deviations. n.s.; not significant.
(TIF)

**S4 Fig. Bar graph of RMM clusters per minute in NEM.** The horizontal axis indicates RMM clusters per minute in NEM and the vertical axis indicates the gestational weeks. The bars represent means and standard deviations. $^*p < 0.05$, n.s.; not significant.
(TIF)

**S5 Fig. Bar graph of MMs per clusters.** The horizontal axis indicates the MMs per cluster and the vertical axis indicates the gestational weeks. The bars represent means and standard deviations. n.s.; not significant.
(TIF)

**S6 Fig. Bar graph of ratio of number of RMMs clusters per minute between NEM and EM periods.** The horizontal axis indicates the ratio of number of RMM clusters per minute between NEM and EM periods and the vertical axis indicates the gestational weeks. The bars represent means and standard deviations. n.s.; not significant.
(TIF)

## Author Contributions

**Conceptualization:** Seiichi Morokuma.

**Formal analysis:** Kana Maehara.

**Funding acquisition:** Seiichi Morokuma.

**Investigation:** Kana Maehara, Kazushige Nakahara, Hikohiro Okawa.

**Project administration:** Seiichi Morokuma.

**Writing – original draft:** Kana Maehara.

**Writing – review & editing:** Seiichi Morokuma, Kiyoko Kato.

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
