## [Decision Letter · Decision Letter 0]

25 Feb 2020

PONE-D-19-33770

A study on the association between eye movements and regular mouthing movements (RMMs) in normal fetuses between 24 to 39 weeks of gestation

PLOS ONE

Dear Morokuma,

Thank you for submitting your manuscript to PLOS ONE. After careful consideration, we feel that it has merit but does not fully meet PLOS ONE’s publication criteria as it currently stands. Therefore, we invite you to submit a revised version of the manuscript that addresses the points raised during the review process.

We would appreciate receiving your revised manuscript by Apr 10 2020 11:59PM. To enhance the reproducibility of your results, we recommend that if applicable you deposit your laboratory protocols in protocols.io, where a protocol can be assigned its own identifier (DOI) such that it can be cited independently in the future. For instructions see: http://journals.plos.org/plosone/s/submission-guidelines#loc-laboratory-protocols

We look forward to receiving your revised manuscript.

Kind regards,

Mehmet Yekta Oncel, M.D.

Academic Editor

PLOS ONE

Additional Editor Comments:

It seems to be interesting, and may have information for readers of the Plos One. My concern about this manuscript is the subjective assessments of mouthing movement and eye movement in this study. Authors should clarify this subject. 

Journal Requirements:

Reviewers' comments:

Reviewer's Responses to Questions

Reviewer #1: General comments

This paper seems to be interesting, and may have information for readers of the PROS ONE. However, I think that significant major revisions are needed to accept this manuscript in the PROS ONE. My big concern is the subjective assessments of mouthing movement and eye movement in this study. How about the inter-observer reproducibility? Authors should address this issue. Authors should clearly state the new points in this study. Authors should also explain the cause for difference in fetal mouthing movements between their study and previous studies.

Specific comments

1. How did fetal movements affect the 2D sonographic observations of the fetal face? I think that authors could not observe fetal face during fetal movements.

2. Authors should state that they used two-dimensional sonography.

3. How about the frequency of the probe.

Reviewer #2: There is a minor typo in the abstract (line 24; 'sacking').

This is a well written manuscript with a clear and concise aim. I do feel that there likely to be a rather loose association between the value of sucking movements as a neurological marker, and that there is too much emphasis placed on the (temporal) association with cortical or synaptic maturation.

There is considerable data emerging on behaviour patterns that involve a much greater integration of the neurological axis, such as the so-called 'General Movements' or behavioural state analysis, which themselves are not the most robust marker of neurological integrity.

Nevertheless, a detailed analysis of individual behavioural parameters as presented in this study certainly do have their place in the literature. I commend the authors for their diligent work.

Reviewer #3: The authors report their experience with evaluation of eye and regular mouthing movements in 2nd and 3rd trimester. They have previously published another article on the same topic. My major concern regards the sonographic methodology employed. In particular, there are at least two issues which should be clarified. 1) reading the paper, it seems that there was no movement whatsoever of any fetus leading to failure of recording the eye and mouth movements in the whole series; 2) in advanced gestation it is rather unusual for the fetuses to lie in such a position that the reference view employed in the study - ie coronal view and contemporary observation of lips and eyes (lenses) - is visible. These two factors should be clarified and explained. From my experience, it is virtually impossible that over 60 uninterrupted minutes of observation the fetuses do not show at least some breathing or major trunk (or limb) movements which will hamper or at least interrupt the continuous observation of the fetal face. And the authors have neither reported any failure in the recording nor acknowledged this in the limitations to the study. furthermore, the single paper (published by the same group) cited to account for reproducibilty is cited as ref n.33, but the reference list stops at 31...

Finally, all the figures attached should report the titles of the axes ON the image and not only in the caption.

In summary, I appreciate the long and tiring work of the authors, but I have doubts about the complete absence or reported failures in the recordings.

To improve the manuscript, this major flaw should be addressed in the discussion and in the presentation of data, and a figure /clip illustrating the sonographic approach should be included

Reviewer #4: This study is well-written and appropriate statistics have been performed. However, can the authors make a comment on the pregnant women characteristics including maternal data; presence of preeclampsia, maternal disease, chorioamnionitis and etc. Were the mothers have these kind of problems, or all were heakthy pregnant women?

The authors explained the limitations of the study. Can they comment on the clinical practical use of these data on both antenatal and postnatal evaluation in both fetuses and infants, respectively?

---

## [Author Response · Author response to Decision Letter 0]

25 Mar 2020

Editor: I have incorporated all of your suggestions into my revision. Thank you for your help.

Reviewer 1: I have incorporated all of your suggestions into my revision. Thank you for your help.

Reviewer 2: I have incorporated all of your suggestions into my revision. Thank you for your help.

Reviewer 3: I have incorporated all of your suggestions into my revision. Thank you for your help.

Reviewer 4: I have incorporated all of your suggestions into my revision. Thank you for your help.

---

## [Decision Letter · Decision Letter 1]

8 Apr 2020

PONE-D-19-33770R1

A study on the association between eye movements and regular mouthing movements (RMMs) in normal fetuses between 24 to 39 weeks of gestation

PLOS ONE

Dear Morokuma,

Thank you for submitting your manuscript to PLOS ONE. After careful consideration, we feel that it has merit but does not fully meet PLOS ONE’s publication criteria as it currently stands. Therefore, we invite you to submit a revised version of the manuscript that addresses the points raised during the review process.

We would appreciate receiving your revised manuscript by May 23 2020 11:59PM. To enhance the reproducibility of your results, we recommend that if applicable you deposit your laboratory protocols in protocols.io, where a protocol can be assigned its own identifier (DOI) such that it can be cited independently in the future. For instructions see: http://journals.plos.org/plosone/s/submission-guidelines#loc-laboratory-protocols

We look forward to receiving your revised manuscript.

Kind regards,

Mehmet Yekta Oncel, M.D.

Academic Editor

PLOS ONE

Additional Editor Comments (if provided):

I think the comments of the referees should be reviewed by the authors. There are some unanswered questions.

Reviewers' comments:

Reviewer's Responses to Questions

**Comments to the Author**

1. If the authors have adequately addressed your comments raised in a previous round of review and you feel that this manuscript is now acceptable for publication, you may indicate that here to bypass the “Comments to the Author” section, enter your conflict of interest statement in the “Confidential to Editor” section, and submit your "Accept" recommendation.

2. Is the manuscript technically sound, and do the data support the conclusions?

3. Has the statistical analysis been performed appropriately and rigorously? 

Reviewer #1: Yes

Reviewer #2: Yes

Reviewer #3: Yes

Reviewer #4: Yes

4. Have the authors made all data underlying the findings in their manuscript fully available?

Reviewer #1: Yes

Reviewer #2: No

Reviewer #3: Yes

Reviewer #4: Yes

5. Is the manuscript presented in an intelligible fashion and written in standard English?

Reviewer #1: Yes

Reviewer #2: Yes

Reviewer #3: Yes

Reviewer #4: Yes

6. Review Comments to the Author

Reviewer #1: General comments

The revised manuscript is significantly improved. However, I think that one minor revision is needed to accept this manuscript in the PROS ONE.

Specific comments

1. The frequency of the probe means 3-MHz, 3,5-MHz, or 5-MHz, etc. Authors should state the frequency of the probe in the text.

Reviewer #2: The authors have made a good effort to address the comments from all the reviewers, and I am happy to support the publication

Reviewer #3: The authors have submitted a new version of the manuscript, apparently revised according to the comments of the reviewers which they claim were "fully considered and included in the revised version". However, at least for the points I raised, they only reported a very generic sentence "Cases where such observation was not possible or time during which observation was not possible were not included". The authors did neither mention the number of cases in which the recording had started and then aborted nor their incidence by weeks of pregnancy. In addition, they did not include a sonographic image to show the technique, nor a clip to this extent.

Since the former was my major criticism, because it is quite clear tha major head and trunk movements do occur quite frequently in the 3rd trimester of pregnancy, I do not think my comments were adequately dealt for. Therefore, I would recommend rejection of the paper, due to possible selection bias and failure to comply with reviewers' suggestions

Reviewer #4: Thank you for the revisions

The authors made all the changes that have been recommended by the reviewers

7. PLOS authors have the option to publish the peer review history of their article (what does this mean?). If published, this will include your full peer review and any attached files.

---

## [Author Response · Author response to Decision Letter 1]

30 Apr 2020

Response to Reviewers

We would like to thank the reviewers for taking the time to review our manuscript and for their useful comments. The reviewers' comments are in italics below, while our responses are typed in bold. Changes in response to the comments have been highlighted in red font in the revised manuscript.

6. Review Comments to the Author

Reviewer #1: General comments

The revised manuscript is significantly improved. However, I think that one minor revision is needed to accept this manuscript in the PROS ONE.

Specific comments

1. The frequency of the probe means 3-MHz, 3,5-MHz, or 5-MHz, etc. Authors should state the frequency of the probe in the text.

Response: Thank you for your kind comments. The frequency of the probe used is 3.5MHz. We have added the frequency of the probe in the text. (Line 100-101)

Reviewer #2: The authors have made a good effort to address the comments from all the reviewers, and I am happy to support the publication

Response: Thank you for this kind comment; we appreciate your support.

Reviewer #3: The authors have submitted a new version of the manuscript, apparently revised according to the comments of the reviewers which they claim were "fully considered and included in the revised version". However, at least for the points I raised, they only reported a very generic sentence "Cases where such observation was not possible or time during which observation was not possible were not included". The authors did neither mention the number of cases in which the recording had started and then aborted nor their incidence by weeks of pregnancy. In addition, they did not include a sonographic image to show the technique, nor a clip to this extent.

Since the former was my major criticism, because it is quite clear tha major head and trunk movements do occur quite frequently in the 3rd trimester of pregnancy, I do not think my comments were adequately dealt for. Therefore, I would recommend rejection of the paper, due to possible selection bias and failure to comply with reviewers' suggestions

Response: Thank you for your comment. We apologize for not fully answering your concerns in the last round of revision. In two cases, the effective observation time was insufficient due to fetal posture and movement; the gestational weeks were 36 and 38 weeks, respectively, when the two cases were observed. We apologize for not including these two examples in the fetal population. Therefore, we have revised the manuscript according to your comment (line 176-178). When we were unable to render the appropriate cross-section, we adjusted the position of the probe and asked the mother to change her position so that the proper cross-section could be rendered. We have revised the manuscript to clarify this (line 105-107). We have also described this as a limitation of the study (line 286-287).

In addition, we have added a figure /clip illustrating the sonographic approach as S1 Fig. Finally, many other researchers have used this research method. Reference number 5, 9, and 16 use a similar research method to the one used in our study. 

Reviewer #4: Thank you for the revisions

The authors made all the changes that have been recommended by the reviewers

Response: Thank you for this kind comment; we appreciate your support.

---

## [Decision Letter · Decision Letter 2]

15 May 2020

A study on the association between eye movements and regular mouthing movements (RMMs) in normal fetuses between 24 to 39 weeks of gestation

PONE-D-19-33770R2

Dear Dr. Morokuma,

We are pleased to inform you that your manuscript has been judged scientifically suitable for publication and will be formally accepted for publication once it complies with all outstanding technical requirements.

With kind regards,

Mehmet Yekta Oncel, M.D.

Guest Editor

PLOS ONE

Additional Editor Comments (optional):

The authors made all the changes that have been recommended by the all reviewers.

Reviewers' comments:

Reviewer's Responses to Questions

Reviewer #3: Now eventually also my comments have been addressed. Thanks. In particular, a picture showing the reference view of the fetal face has been added. And the number of cases discarded because of fetal movements too, even though in my experience these should have been much higher...

---

## [Editor Report · Acceptance letter]

19 May 2020

PONE-D-19-33770R2 

A study on the association between eye movements and regular mouthing movements (RMMs) in normal fetuses between 24 to 39 weeks of gestation 

Dear Dr. Morokuma:

I am pleased to inform you that your manuscript has been deemed suitable for publication in PLOS ONE. Congratulations! Your manuscript is now with our production department. 

With kind regards,

on behalf of

Dr. Mehmet Yekta Oncel 

Guest Editor

PLOS ONE